climatology/environmental science

nitrogen, nitrate losses, catchment, agriculture, Norway, climate change

**Author for correspondence:**
Marianne Bechmann
e-mail: marianne.bechmann@nibio.no

# Nitrogen losses from two contrasting agricultural catchments in Norway

Xueli Chen[1,2] and Marianne Bechmann[3]

[1]Institute of Soil Fertilizer and Environment Resource, Heilongjiang Academy of Agricultural Sciences, Harbin 150086, People's Republic of China
[2]Heilongjiang Academy of Agricultural Sciences Postdoctoral Programme, Harbin 150086, People's Republic of China
[3]Norwegian Institute of Bioeconomy Research, PO Box 115, 1431 Ås, Norway

 XC, 0000-0002-2408-1269; MB, 0000-0002-3937-4582

Nitrogen (N) losses from agricultural areas, especially into drinking water and marine environments, attract substantial attention from governments and scientists. This study analysed nitrogen loss from runoff water using long-term monitoring data (1994–2016) from the Skuterud catchment in southeastern Norway and the Naurstad catchment in northern Norway. Precipitation and runoff were lower in the Skuterud catchment than in the Naurstad catchment. However, in the Skuterud catchment, the annual total N (TN) losses ranged from 27 to 68 kg hm$^{-2}$. High precipitation (1247 mm) in the Naurstad catchment resulted in substantial runoff water (1108 mm) but relatively low total TN losses ranged from 17 to 35 kg hm$^{-2}$. The proportion of nitrate losses to TN loss was 51–86% and 28–50% in the Skuterud and Naurstad catchments, respectively. Furthermore, the monthly average TN concentrations and nitrate losses had two peaks, in April–May and October, in the Skuterud catchment; however, no significant fluctuations were found in the Naurstad catchment. The contributions of N and runoff water to TN and nitrate losses were calculated using multiple linear regression, and runoff water was the major contributor to TN loss in both catchments. Runoff water was the main factor in the Skuterud catchment, and the nitrate-N concentration was the main factor in the Naurstad catchment.

## 1. Background

Nitrogen (N) is an essential nutrient for maintaining crop yields and meeting food requirements for the increasing global population. However, during the last century, human contributions to the N cycle began to increase dramatically. For instance, in China, N fertilizer consumption reached 23.1 million tons (Mt) in 2016, which increased approximately 50%

**Table 1.** Characteristics of the two catchments.

| catchment | total area (hm$^2$) | agricultural land use (%) | main crops | livestock units (hm$^{-2}$) | soil status (TN mg kg$^{-1}$) | N application (kg hm$^{-2}$)[a] | monitoring period |
|---|---|---|---|---|---|---|---|
| Skuterud | 449 | 61 | cereals | 0.257 | 16.00 | 162 | 1994–2016 |
| Naurstad | 146 | 35 | grass | 0.840 | 22.04 | 112 | 1994–2016 |

[a]Mean annual nitrogen fertilizer application (kg hm$^{-2}$) for the monitoring period (1994–2016).

compared with consumption in 1990 [1]. In Norway, the total consumption of N was approximately 0.1 Mt in 2014, corresponding to approximately 100 kg N hm$^{-2}$ arable land [2]. Increases in N fertilizer inputs and the N surplus result in the leaching of N from agricultural soils, and N losses at the catchment scale result in environmental problems in streams, rivers, lakes and the sea [3–5]. According to Food and Agriculture Organization (FAO) projections in 'World Agriculture: Towards 2015/2030' [6], N losses will grow substantially in the world's increasingly intensive agricultural systems, despite improvements in N use efficiency [7].

The levels of total N and nitrate-N losses from agricultural areas in Nordic and Baltic countries have varied from 6.6 to 100 kg hm$^{-2}$ and from 4 to 76 kg hm$^{-2}$, respectively [8]. From 1989 to 1996, in Denmark, there was a large decrease in N losses; the average amount of nitrate-N that leached from sandy catchments was 123 kg N hm$^{-2}$ yr$^{-1}$, and the average amount from loamy catchments was 72 kg N hm$^{-2}$ yr$^{-1}$ [9], whereas the long-term average was 16–47 kg hm$^{-2}$ in Norway [8]. In Sweden, annual N leaching losses from arable land in south Sweden usually range from 15 to 45 kg hm$^{-2}$ [10].

Soil tillage has been shown to increase N losses in agricultural soils due to increased mineralization and the absence of plant N uptake [11–13]. Furthermore, in Norway, the relatively high concentrations of nitrate-N in drainage water from fallow plots resulted from the large amounts of nitrate-N mineralized in the soil during the summer season [14]. Studies have also reported that the linear correlation between N leaching in autumn and N surface balance was good ($R^2 = 0.95$) in the Skuterud catchment during years with similar weather conditions in Norway [15]. The annual losses of N from 1994 to 2006 were 13–15 kg N hm$^{-2}$ and 7–17 kg N hm$^{-2}$ at the Skuterud catchment and Naurstad catchment, respectively [16]. However, the seasonal change in N loss and the driving factors remain unclear. In this study, we selected contrasting catchments, the Skuterud and Naurstad catchments, to study the effects of differences in soil types (silty soil and peat), crop types (cereal and grassland), climate and animal stocking rates on the N surplus during production, N concentrations and losses, the seasonality in N losses, and the forms of N lost from the two contrasting systems. We assume that different soils, land uses, climate and farming management systems may potentially have a significant influence on N dynamics. In this paper, total N and nitrate-N losses and concentrations in runoff water in Skuterud and Naurstad catchments, two representative areas in Norway, were analysed based on 23-year monitoring data (1994–2016). The objectives of this study were to address knowledge gaps related to N in runoff water and the links with agricultural practices.

# 2. Methods

## 2.1. Site description

The Skuterud and Naurstad catchments are parts of the National Environmental Monitoring Programme (JOVA). The Skuterud catchment is located in southeastern Norway and is 91–146 m.a.s.l. It has a total area of 4.5 km$^2$, of which approximately 61% (2.7 km$^2$) is agricultural land, and the remaining area includes forests, inhabited areas, bogs and roads. There are 51 farm fields within this catchment, and the land is used primarily for cereal production (table 1). All fields are systematically tile drained. Farming operations at the field level (e.g. time and rate of fertilizer and manure application, tillage, livestock, sowing and yields) have been recorded every year since the start of water quality monitoring in 1994. The soils primarily originate from marine deposits. The agricultural land is mostly covered by clay loam, marine silt loam and silty clay loam soils (table 2). Coarser marine beach deposits with loamy sandy texture predominate on the fringes of agricultural lands near and in the forested area. The main

**Table 2.** Weather and elevation characteristics.

| catchment | soil texture | standard 30-year-normal period | | monitoring period | | elevation range (m.a.s.l.) | length of growing season (days) |
| | | mean annual temperature (°C)[a] | mean annual precipitation (mm)[a] | mean annual temperature (°C)[b] | mean annual precipitation (mm)[b] | | |
| --- | --- | --- | --- | --- | --- | --- | --- |
| Skuterud | clay loam, silt loam | 5.5 | 785 | 6.2 | 909 | 91–146 | 194 |
| Naurstad | peat soil on fine sand | 4.5 | 1020 | 5.2 | 1247 | 5–75 | 173 |

[a]1961–1990; [b]1994–2016.

reference soil groups of the arable land are classified as Epistagnic Albeluvisols (Siltic), Luvic Stagnosols and Endostagnic Cambisol (Dystric) (World Reference Base for soil resources, WRB). In southeastern Norway, there are predominately clay soils of marine origin; thus, the Skuterud catchment can be regarded as representative of this region with respect to soils. The climate is continental with cold winters and relatively warm summers. The growing season lasts for 194 days.

The Naurstad catchment is located in northern Norway. It has a total area of 1.5 km$^2$, of which approximately 35% (0.63 km$^2$) is agricultural land, and the remaining areas are forests, inhabited areas, bogs and roads (table 1). There are 18 farm fields within the catchment, and the land is used primarily for grass production. All fields are systematically tile drained, and they vary in size from 0.6 to 3.8 hm$^2$ with a mean of approximately 1.6 hm$^2$. Farming operations at the field level (e.g. time and the rate of fertilizer and manure application, tillage, sowing and yields) have been registered since the start of water quality monitoring in 1994. The catchment contains a flat area and hills close to the sea coast (4–91 m.a.s.l.) and is surrounded by mountains. Most of the agricultural area is drained bog land, where the soil consists of a peat layer covering sand originating from marine shore deposits (table 2). In the Naurstad catchment, there is a coastal climate with relatively mild winters and high precipitation rates during summers, and the growing season lasts for 173 days.

## 2.2. Monitoring and analysis methods

The monitoring data contained time-series information on agricultural practices within the catchment and continuous measurements of runoff and water quality in the main stream. The information on agricultural practices was derived from annual questionnaires to farmers regarding their management of each field.

In both streams of the two catchments, water levels were recorded automatically using a pressure transducer in combination with a data logger (Campbell Scientific). Discharges were calculated based on the existing head–discharge relation. Water samples were collected automatically on a volumetric-proportional basis at the monitoring stations [17]. Water sampling was implemented by the data logger when a fixed predetermined volume of water had past the measuring station, while a small water sample was taken and added to a container that was kept in a refrigerator in the monitoring station. Samples were collected for analysis every 14 days.

The collected samples were analysed for total N (TN) and nitrate-N content. TN was determined by persulfate digestion followed by analysis with a spectrophotometer (Norwegian standard NS-ENISO 13395). Nitrate-N was determined according to a Norwegian standard method (EPA-600/4-79-020). Annual loads were calculated as the cumulative sum of hourly loads derived from hourly flow values and fortnightly concentration values. Average concentrations were calculated on an annual basis and weighed according to runoff volume.

## 2.3. Calculations and statistics

The annual runoff, TN loss and nitrate-N loss from the two catchments were calculated as the sum of each sampling period within a year. Monthly runoff and loss of TN and nitrate-N were presented by

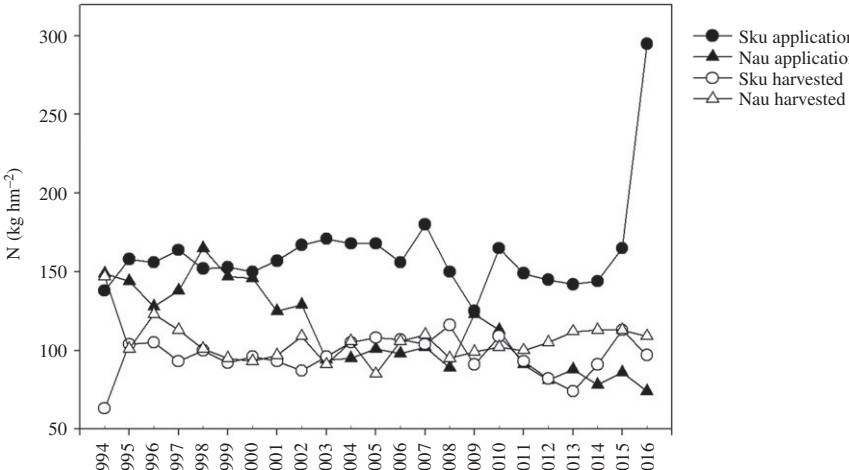

**Figure 1.** Nitrogen application and N harvested in the Skuterud and Naurstad catchments from 1994 to 2016. Sku and Nau represent the Skuterud and Naurstad catchments, respectively.

the average value each month during the monitoring period. In this paper, organic N (ON) loss was calculated as follows:

$$ON = TN\,(kg\,h\,m^{-2}) - nitrate\text{-}N.$$

where organic N (ON) means nitrogen lost from runoff water in organic nitrogen form (kg hm$^{-2}$), total N (TN) means nitrogen lost from runoff water in total nitrogen form (kg hm$^{-2}$), and nitrate-N (NN) means nitrogen lost from runoff water in nitrate nitrogen form (kg hm$^{-2}$).

The annual concentrations of TN and nitrate-N were calculated as the average of each sampling period within a year. Monthly concentrations of TN and nitrate-N were represented by the average value each month. Statistical analysis was performed by using SPSS v. 22.0 software, and figures were drawn using SigmaPlot v. 10.0 software. The contributions of concentration and runoff water to N loss were analysed in R [18], and each variable was transformed to have a mean of 0 and a standard deviation of 1. Multiple linear regression, the *lm()* function in R [18], was performed to study how N concentration and runoff water contributed to total and nitrate-N losses based on annual data.

# 3. Results

## 3.1. Nitrogen application and N harvested in two catchments during the monitoring period

From 1994 to 2016, the average N applications in the Skuterud and Naurstad catchments were 162 and 112 kg hm$^{-2}$, respectively (figure 1). In the Skuterud catchment, the average annual N application during the monitoring period varied from 125 to 295 kg hm$^{-2}$ in 2016. The high amount of N input in 2016 was mainly from sewage sludge application according to the farmer questionnaires. In the Naurstad catchment, N application was generally low and ranged from 125 to 165 kg hm$^{-2}$ from 1994 to 2002, and declined to 74–112 kg hm$^{-2}$ from 2003 to 2016; however, the application was 123 kg hm$^{-2}$ in 2009, which was similar to the application amounts in the previous period. During the monitoring period, there was a significant decrease in N application in the Naurstad catchment.

Nitrogen harvested in the catchments resulted in the removal of N from fields with crops (figure 1). The average annual amount of N harvested from the two catchments fluctuated between 74 and 116 kg hm$^{-2}$, except in 1994, when the lowest amount of N harvesting occurred in Skuterud (63 kg hm$^{-2}$) and the highest amount of N harvesting occurred in Naurstad (147 kg hm$^{-2}$). The average N harvested in the two catchments was 96 and 105 kg hm$^{-2}$ for Skuterud and Naurstad, respectively.

The average N surplus in production over the monitoring period was 66 kg hm$^{-2}$ for cereal production in the Skuterud catchment and only 7 kg hm$^{-2}$ for grass production in the Naurstad catchment.

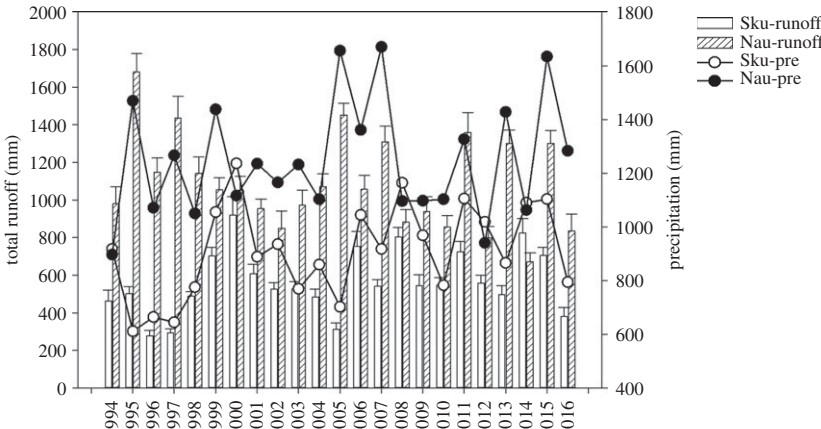

**Figure 2.** Runoff and precipitation in the Skuterud and Naurstad catchments from 1994 to 2016. Sku-pre and Nau-pre represent precipitation in Skuterud and Naurstad catchments, respectively.

**Table 3.** Average concentrations and losses of TN and nitrate-N in the Skuterud and Naurstad catchments.

| catchment | average annual concentration of TN (mg l$^{-1}$) | average annual concentration of nitrate-N (mg l$^{-1}$) | average losses of TN (kg hm$^{-2}$) | average losses of nitrate-N (kg hm$^{-2}$) |
|---|---|---|---|---|
| Skuterud | 5.51 | 4.29 | 42.4 | 32.3 |
| Naurstad | 1.1 | 0.38 | 24.3 | 8.7 |

## 3.2. A high amount of precipitation results in a high amount runoff water

The average annual precipitation values in Skuterud and Naurstad were 909 and 1247 mm, respectively, during the monitoring period (1994–2016) (figure 2), and these annual precipitation values were higher than those that occurred in the 30-year-normal period (1961–1990), as presented in table 2. The average annual temperatures in the Skuterud and Naurstad catchments during the monitoring period were 6.2°C and 5.2°C, respectively, which were higher than the 30-year average annual temperature values 5.5°C and 4.5°C, respectively (table 2). In comparison with the 30-year-normal (1961–1990) annual precipitation and average temperature, the monitoring period was generally warmer and wetter.

The average amounts of runoff water for the Skuterud and Naurstad catchments were 546 and 1108 mm, respectively. The results showed that the runoff water varied from 278 to 919 mm yr$^{-1}$ in Skuterud and from 671 to 1681 mm yr$^{-1}$ in Naurstad. Comparing the two catchments, precipitation and runoff were much higher in the Naurstad catchment than in the Skuterud catchment, except in 2000 and 2008, when the catchments had similar amounts of runoff and precipitation (figure 2).

## 3.3. Annual nitrogen concentration

The average annual TN concentrations in Skuterud and Naurstad were 5.5 and 1.1 mg l$^{-1}$, respectively (table 3). Correspondingly, the annual average concentrations of nitrate-N were 4.3 and 0.4 mg l$^{-1}$ in Skuterud and Naurstad, respectively. The fluctuations in TN and nitrate-N concentrations were similar between the two streams (figure 3). In the Skuterud catchment, the annual TN and nitrate-N concentrations during the monitoring period varied from 3.7 to 7.5 mg l$^{-1}$ and from 2.6 to 6.5 mg l$^{-1}$, respectively. In 2001, in the Skuterud stream, the concentrations of TN and nitrate-N were 3.7 and 2.6 mg l$^{-1}$, respectively, which were lower than in other years. Then, the concentrations of both TN and nitrate-N in Naurstad were stable.

## 3.4. Monthly average nitrogen concentration

The monthly average N concentration showed significant differences throughout the year between the two catchments. In the Skuterud catchment, there were two peaks for the monthly average concentrations of both TN and nitrate-N during the year (figure 4). The first peak appeared, in May,

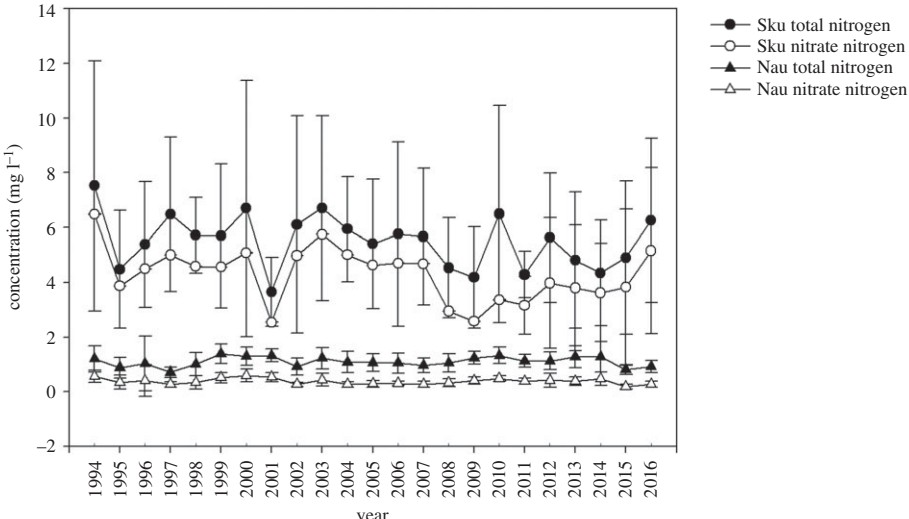

**Figure 3.** Total N and nitrate-N concentrations in the Skuterud and Naurstad catchments from 1994 to 2016. Bars show minimum and maximum variations in nitrogen concentrations. Sku and Nau represent the Skuterud and Naurstad catchments, respectively.

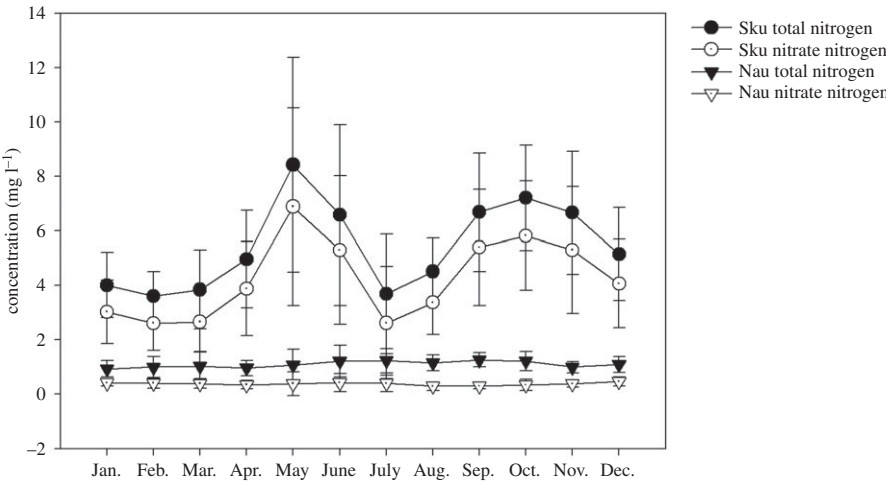

**Figure 4.** Monthly average values for TN and nitrate-N concentrations in 1994–2016 in the Skuterud and Naurstad catchments. Sku and Nau represent the Skuterud and Naurstad catchments, respectively.

at 8.4 mg l$^{-1}$ for TN and 6.9 mg l$^{-1}$ for nitrate-N. Then, the concentration decreased to a minimum level in July. In October, the TN and nitrate-N concentrations reached second peaks of 7.2 and 5.8 mg l$^{-1}$, respectively; then, the concentrations decreased again to low levels from February to March (figure 4). In the Naurstad catchment, neither TN nor nitrate-N concentrations showed significant variations (figure 4).

## 3.5. Annual nitrogen loss in the two catchments

The average total N losses from the Skuterud and Naurstad catchments were 42.4 and 24.3 kg hm$^{-2}$, respectively, while the annual nitrate-N losses from the two catchments were 32.2 and 8.7 kg hm$^{-2}$, respectively (figure 5). TN loss varied each year in the Skuterud catchment; TN loss tended to decrease in the Naurstad catchment, but the decrease was not statistically significant ($0.05 < p < 0.1$) [19] during the monitoring period. In 2000 and 2008, the amounts of runoff water were similar between the two catchments (figure 2), but TN concentration and TN loss were much higher in the Skuterud catchment than in the Naurstad catchment (figures 3 and 4). Nitrate-N loss accounted for 50.9–87.4% and 25.96–50.9% of the TN losses in the Skuterud and Naurstad catchments, respectively (figure 5). There was a significant difference in the loss of nitrate-N between the two catchments.

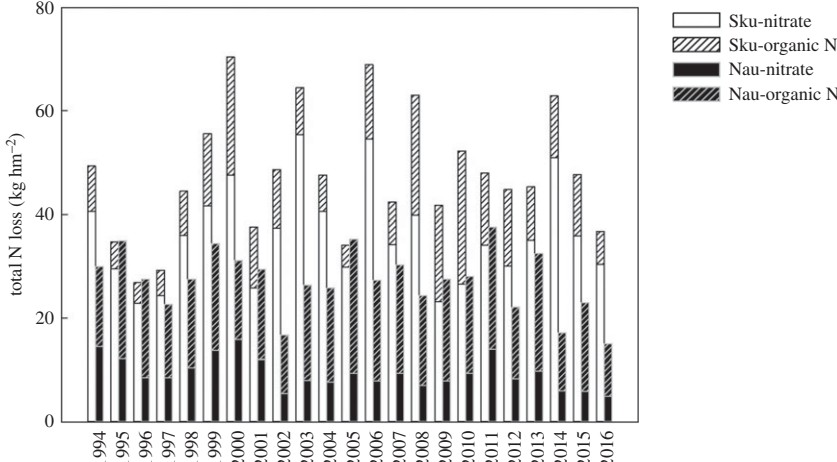

**Figure 5.** Nitrogen losses (Sku-nitrate, Sku-organic N, Nau-nitrate and Nau-organic N) in the two catchments from 1994 to 2016. Sku and Nau represent the Skuterud and Naurstad catchments, respectively.

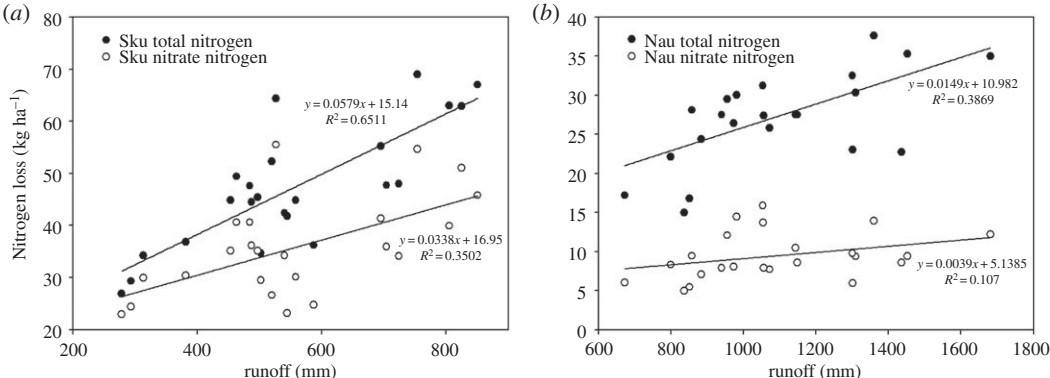

**Figure 6.** Correlation between monthly N loss and runoff in 1994–2016. (*a*) Skuterud catchment; (*b*) Naurstad catchment.

Correspondingly, the loss of organic N varied between 4.0 and 25.7 kg hm$^{-2}$ in the Skuterud catchment and between 11.2 and 25.9 kg hm$^{-2}$ in the Naurstad catchment, accounting for 14.7–49.1% and 49.1–74.1% of the TN loss, respectively. Comparing the amounts of organic N loss from the two catchments, the difference was not significant for most monitoring years. This result means that the organic N produced was not affected by different agricultural systems and was relatively stable over the long-term period (figure 5).

## 3.6. Correlation between runoff and nitrogen losses

A strong correlation was found between the annual TN loss and runoff in both the Skuterud and Naurstad catchments from 1994 to 2016. The correlation coefficients were $r^2 = 0.65$ for Skuterud and $r^2 = 0.38$ for Naurstad. The results indicated that increased runoff yielded greater TN losses (figure 6). A correspondence analysis was conducted for nitrate-N losses. The relative coefficient was 0.35 in the Skuterud catchment, which demonstrated that the amounts of N loss in the form of nitrate-N exhibited a strong relationship with runoff in Skuterud. This value was 0.10 in the Naurstad catchment because nitrate-N was not the main form in the runoff water.

## 3.7. Monthly average nitrogen loss and runoff

The results of monthly runoff water showed that there were two peaks throughout the year, one in April in both catchments and the other in November for the Skuterud catchment and in October for the Naurstad catchment, with an average value of 144.5 and 81.1 mm, respectively, throughout the monitoring period (figure 7). When comparing the two catchments, the amount of runoff water

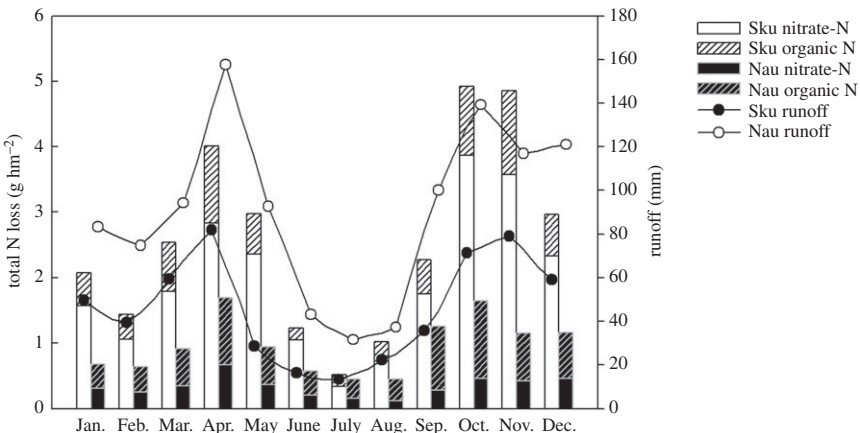

**Figure 7.** Monthly N losses and runoff during 1994–2016. In this figure, nitrate-N + organic N = total N. Sku and Nau represent the Skuterud and Naurstad catchments, respectively.

**Table 4.** Calculations of concentrations and runoff water contributions to N loss.

| site | formulae | notes |
|------|----------|-------|
| Skuterud | $TN_{Sku} = 0.45 \times C_{TN} + 0.84 \times RW_{Sku} - 9.573 \times 10^{-17}$ | $TN_{Sku}$, total nitrogen loss, kg hm$^{-2}$; $C_{TN}$, total nitrogen concentration of runoff water, mg l$^{-1}$; $RW_{Sku}$, amounts of runoff water, mm |
| | $NN_{Sku} = 0.78 \times C_{NN} + 1.59 \times RW_{Sku} + 0.925$ | $NN_{Sku}$, nitrate-N loss, kg hm$^{-2}$; $C_{NN}$, nitrate-N concentration of runoff water, mg l$^{-1}$; $RW_{Sku}$, amounts of runoff water, mm |
| Naurstad | $TN_{Nau} = 0.74 \times C_{TN} + 1.14 \times RW_{Nau} + 0.587$ | $TN_{Nau}$, total nitrogen loss, kg hm$^{-2}$; $C_{TN}$, total nitrogen concentration of runoff water in Naurstad, mg l$^{-1}$; $RW_{Nau}$, amounts of runoff water in Naurstad, mm |
| | $NN_{Nau} = 2.07 \times C_{NN} + 0.35 \times RW_{Nau} + 0.882$ | $NN_{Nau}$, nitrate-N loss, kg hm$^{-2}$; $C_{NN}$, nitrate-N concentration of runoff water, mg l$^{-1}$; $RW_{Nau}$, amounts of runoff water, mm |

was higher in Naurstad than in Skuterud in all months. Runoff was approximately 81 and 65 mm higher in September and October, respectively, in Naurstad than in Skuterud. The results showed that TN losses mainly occurred in April, October and November in both catchments during the monitoring period, indicating the same trends as the runoff (figure 7). Although there was a peak in TN concentration in May in the Skuterud catchment (figure 4), this peak did not result in high TN loss. The TN loss was higher in the Skuterud catchment than in the Naurstad catchment in all months, except in July and September, when TN loss showed an opposite trend to the amount of runoff water in the two catchments. The monthly average nitrate-N loss during the monitoring period in the two catchments also showed two peaks, with one in April and one in October. The differences occurred in the percentages of nitrate-N loss as a part of the TN loss, which for monthly losses were 63.7–83.2% in the Skuterud catchment and 21.7–43.1% in the Naurstad catchment.

## 3.8. Contributions of concentration and runoff to nitrogen loss

Runoff water was the main factor that determined the TN loss in both the Skuterud and Naurstad catchments, with coefficients of 0.93 and 0.99, respectively. TN concentrations had coefficients of only 0.42 and 0.79 for the Skuterud and Naurstad catchments, respectively (table 4); however, the two predictors, runoff water and TN concentration, behaved differently in terms of nitrate-N loss, as runoff water was the main factor in Skuterud and N concentration was the main factor in Naurstad (table 4).

# 4. Discussion

In comparison with the Naurstad catchment, which is dominated by grasslands, the Skuterud catchment in southern Norway has a highly intensive agricultural cropping system with mainly cereal production. The N surplus in the soil, which is defined as the difference between the N applied and the N harvested, may increase the risk of N leaching [20]. The average N surplus was 66 kg hm$^{-2}$ in the Skuterud catchment and 7 kg hm$^{-2}$ in the Naurstad catchment over the monitoring period. The difference was highest from 2000 to 2010 (figure 1), when N application in the Naurstad catchment decreased. The longer growing season for grass in Naurstad compared with the cereal growing season in the Skuterud catchment resulted in the increased uptake of N and hence the low N surplus in Naurstad. It has been shown that the length of the growing season significantly influences the uptake of N and hence the N surplus [21].

The study results showed the agricultural area in the Skuterud catchment (61%, 273.9 hm$^2$) was more than five times the agricultural area in Naurstad (35%, 51.1 hm$^2$), and the average annual TN concentration in Skuterud was more than five times that in Naurstad. The nitrate-N concentration in the Skuterud catchment was approximately 10 times that in the Naurstad catchment (table 3). It has been shown that the TN concentration is significantly correlated with the per cent of agricultural area [22]. The annual concentrations of N in runoff water in our study were consistent with the results of previous studies for 1994–2006. Previous studies documented that the annual mean concentration in the Skuterud catchment was 6.0 mg TN l$^{-1}$, and the lowest annual TN concentration (1.1 mg TN l$^{-1}$) was observed in the grassland-dominated Naurstad catchment in northern Norway [16].

Agricultural management systems affect N use efficiency, which can be improved by improved nutrient management, e.g. cropping systems [23]. Additionally, riparian buffer zones and constructed wetlands may increase the retention of N in the catchment by increasing the N loss to air (denitrification) and by harvesting N in biomass [24]. These changes may contribute to the reduction in N losses and the development of sustainable agriculture [23]. Furthermore, N losses from grasslands have been shown to be much lower than N losses from areas with soil tillage and cereal crops [25,26]. Nitrogen leaching was affected not only by tillage practices and ploughing time in spring or in autumn but also by soil texture [27,28]. N mineralization is determined by the rate of total carbon to total nitrogen in a silty clay soil [29] and by substrate characteristics in addition to temperature and hydrology in peat soils, such as in the Naurstad catchment [30]. Silty clay is the dominant soil type in the Skuterud catchment and peat is the dominant soil type in Naurstad. In comparison with the Naurstad catchment, the Skuterud catchment has more soil tillage, a longer period without crop cover and a warm climate, which may contribute to the differences in the TN losses between the catchments [24].

Nitrate-N was found to be the main type of N loss from agricultural land-use catchments [31]. Generally, the amounts of nitrate-N in runoff water are influenced by N mineralization, local precipitation, crop systems [32] and discharge [13], and soil freeze–thaw cycles during winter could be another dominant factor leading to high N losses and N concentrations in runoff water [33]. In this study, higher losses of nitrate-N were found in the Skuterud catchment than in the Naurstad catchment (figure 4), which may be due to the high temperature, more soil tillage, high N fertilizer application and N surplus in Skuterud, and therefore increased N mineralization [34]. Nitrogen mineralization occurring due to N leaching increases after ploughing, in periods without plant growth and when temperatures are still high [10], and 57% of the N loss occurred from the catchment during the period between after harvest and before spring planting when crops were not present [32]. Such a periodic variation is well known in areas dominated by annual crops [22]. Nitrogen loads in agriculture-dominant catchments depend on seasonal conditions [13]. Owing to increased uptake by plants, the lowest N concentrations were observed during the summer months (June–August) [5,10]. Although N mineralization accelerated under high temperatures during May to August, the utilization of N for crop growth also increased during this period, which decreased the risk of nitrate-N losses. Then, with the crops harvested, mineralized N accumulated in the soil, and high nitrate-N losses occurred under heavy precipitation from September to December. These results were consistent with those of Povilaitis *et al.* [5], who also found that the TN concentration showed a remarkable increase in November and December due to the higher amounts of nitrate-N available in the soil once uptake by vegetation ceases. In the Naurstad catchment, the TN concentrations were low throughout the year and varied between 0.9 mg l$^{-1}$ in April and greater than 1.13 mg l$^{-1}$ from May to October. Organic N was the main form of N loss, which resulted from grass-covered land year-round with low nitrate losses due to plant uptake and organic

matter mineralization in winter. This result was consistent with the results of Kucke & Kleeberg [35]. Peat and sandy soils with high water-infiltration capacity dominate the Naurstad catchment, and probably parts of the tile drains dilute the N concentrations that enter the groundwater. Because of the greater groundwater supply, the water residence time may be prolonged in the Naurstad catchment, and therefore, larger quantities of nitrate-N can be removed by denitrification [5,36]. In the Naurstad catchment, the percentage of nitrate-N in TN loss from December to February of the second year was on average 43%, which was higher than the percentage loss in other months. Nitrogen leaching loss was influenced much more by the amount of organic matter accumulated in the soil in previous years than by the mineral N fertilization of the current crop [35].

# 5. Conclusion

Based on 23 years of continuous monitoring in two catchments located in northern and southeastern Norway, we found that nitrate-N was the main form of N lost in the Skuterud catchment and organic N was the main form of N lost in the Naurstad catchment. Nitrogen concentrations and losses in the Skuterud catchment were much higher than those in the Naurstad catchment. Nitrogen concentrations in the Skuterud catchment showed seasonal variations with the highest concentrations in spring and autumn, whereas the nitrogen concentrations in the Naurstad catchment did not vary much over the year. Continuous plant cover and N uptake are important measures to reduce N losses. Furthermore, the results showed that increased runoff resulted in increased N loss in the Skuterud catchment. Hence, in a future climate with expected increased runoff rates, the need for measures to reduce N loss, e.g. catch crops and split N application, will increase. In the Naurstad catchment, N losses are relatively low and have decreased due to reduced N applications and N surplus throughout the monitoring period.

Data accessibility. Data available from the Dryad Digital Repository at: https://doi.org/10.5061/dryad.j0zpc868v [37].
Authors' contributions. X.C. carried out the statistical analyses and drafted the manuscript. M.B. carried out the study design, collected field data, coordinated the study and helped draft the manuscript. All authors gave final approval for publication.
Competing interests. The authors declare that they have no competing interests.
Funding. This work was supported by grants from the Central Committee's Special Project for Guiding Local Science and Technology Development (grant no. ZY18A04), postdoctoral funding from Heilongjiang Province (grant no. LBH-Z17198) and funding from the Heilongjiang Academy of Agricultural Sciences (grant no. 2018YYYF017).
Acknowledgements. We thank the Agricultural Environmental Monitoring Programme (JOVA) for setting up monitoring stations throughout Norway and all members who contributed to sample collection and daily maintenance.

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
