## [Reviewer comments · Royal Society Open Science]

Review History

RSOS-190490.R0 (Original submission)

Review form: Reviewer 1

Is the manuscript scientifically sound in its present form?

Yes

Are the interpretations and conclusions justified by the results?

Yes

Is the language acceptable?

Yes

Is it clear how to access all supporting data?

Not Applicable

Do you have any ethical concerns with this paper?

No

Have you any concerns about statistical analyses in this paper?

No

Recommendation?

Major revision is needed (please make suggestions in comments)

Comments to the Author(s)

The paper contributes to better understanding of the impact of farming practices to surface water quality in small agriculture dominated catchments.

Page 1, line 57 and page 3 lines 16-17: Perhaps "...13-15 kg hm⁻² and 7-17 kg hm⁻²..." and "declined to 74-112 kg hm⁻².."?

P 3, lines 21-23 : N harvested from the two catchments is not exactly similar (96 and 105 kg hm⁻²).

P3, line 29: "normal period" "normal temperature"? long-term mean in 1961-1990? Long-term mean temperature in 1961-1990?

P3, line 38: Please provide more info about the coefficients. What are they?

P3, lines: 42-23: what does "similar variations" mean? Similar in-stream concentrations or between the streams?

P 3, lines 58-60: The results of N loss in the catchments are provided already in the beginning of chapter 4.4.

P3, last row: perhaps 11.3 kg hm⁻²?

P4: It would be more logical if the chapter 4.6 (monthly concentrations) follows the chapter 4.3 (annual concentrations.)

P4: Something is wrong with numbering of figures 5, 6 and 7. Figure 5 is about monthly average values for TN and nitrate-N concentrations and not about N runoff. Fig 6 is about N loss and not N concentrations and Fig 7 is about correlation between concentrations and water runoff and N loss.

P4, lines 40-41. How to understand provided indexes. Index is a relationship between two numbers. Please add some explanation.

P4, line 49: "...and more intensive agricultural production with annual crops." What does this mean? Please consider revising of the sentence.

P 4, lines 50-52: Please add actual data about N surplus in the results part.

P4, lines 54-55: Please remove the sentence. You already described the used method to calculate N loss in the methods chapter?

P4, line 56: Something is missing here. "The study results showed that....."?

P 4, line 58: "mean annual concentration"?

P5, line 3: How to understand that? N use efficiency and N losses are largely defined by cropping and tillage systems?

P5, lines 5-6: "In this study.....". This information is provided already under the site descriptions.

P5, lines 9-10: dominating soil type..?

P5, line 18: "...increased soil tillage..." . Increased compared to some previous periods? Any data?

P5, lines 20-24: How to understand that? Fall is a typical season without plant growth?

Temperatures are increasing? Lower summer temperatures are responsible for more mild climate? Probably not "climate" but "weather conditions". Climate is defined as long-term (> 30 years) average state of weather condition.

P5, line 30: Should be Povilaitis et al. (33).

P5, lines 39-41: OK. Biochar could prevent N loss. Whether biochar is actually applied in the Naurstad catchment? How does this sentence relates to your work? Please provide some more explanation.

P5, lines 43-44: One of the previous studies revealed that “Nitrogen leaching loss was influenced much more by the amount of organic matter...” How do you know this the case in two studied catchments? “..was probably influenced...”?

P5, line 53: “.... but these properties were not present in the Naurstad catchment”. Is it really so? N concentrations were rather stable in the Naurstad catchment (e.g. Figure 5) but seasonal N loss varied a lot?

P 6: Please check names of the authors for reference 33.

P 8: figure captions. figures 1-4 in 1994-2016, in 1994-2010.

Figure 7 includes two figures (7A and 7B). Which one is A and B? Axis titles and units are missing?

The manuscript can be accepted for publication if these concerns are properly addressed.

Review form: Reviewer 2

Is the manuscript scientifically sound in its present form?

Yes

Are the interpretations and conclusions justified by the results?

Yes

Is the language acceptable?

Yes

Is it clear how to access all supporting data?

Not Applicable

Do you have any ethical concerns with this paper?

No

Have you any concerns about statistical analyses in this paper?

Yes

Recommendation?

Major revision is needed (please make suggestions in comments)

Comments to the Author(s)

The paper is generally well structured and written. However, I have some comments:

What was the frequency of water quality tests?

Line 58 Total Nitrogen is the sum of nitrate (NO₃), nitrite (NO₂), organic nitrogen and ammonia.

The mathematical formula requires a comment, maybe the Authors should change the symbols.

Figure 7 has no descriptions (incorrectly created pdf), I do not know if it contains relevant data and matching measures.

Table 1 - main crops - GRAIN? Maybe cereals?

Table 2 - please remove “normal temperature” and “normal precipitation”. Use “mean annual temperature” annual precipitation, Long-term - please define this period 1994-2016.

It was mentioned about the correlation between runoff and nitrogen losses (Figure 5A does not exist in the pdf file), what about rainfall-runoff correlation (Figure 2 does not contain this

information)? Nitrogen transfer pathways to surface waters should be clearly described/indicated. Section 4.8 is unclear.
Please, emphasize the contribution of your research to the literature.

Review form: Reviewer 3

Is the manuscript scientifically sound in its present form?

Yes

Are the interpretations and conclusions justified by the results?

Yes

Is the language acceptable?

Yes

Is it clear how to access all supporting data?

Yes

Do you have any ethical concerns with this paper?

No

Have you any concerns about statistical analyses in this paper?

Yes

Recommendation?

Accept with minor revision (please list in comments)

Comments to the Author(s)

The paper is interesting reporting on long-term N losses from nordic agricultural catchments. It provides valuable information for researchers and stakeholders. However, a minor revision is recommended.

Regarding the measures to mitigate N losses there is a gap which is world-wide accepted. I would recommend to shortly discuss the role of riparian buffer zones and constructed wetlands. See e.g.: Mander, Ü., Tournebize, J., Tonderski, K., Verhoeven, J.T.A., Mitsch, J.W., 2017.

Planning and establishment principles for constructed wetlands and riparian buffer zones in agricultural catchments. Ecological Engineering 103B, 296-300.

Decision letter (RSOS-190490.R0)

23-Aug-2019

Dear Dr Chen,

The editors assigned to your paper ("Nitrogen losses from two contrasting agricultural catchments in Norway") have now received comments from reviewers. We would like you to revise your paper in accordance with the referee and Associate Editor suggestions which can be

found below (not including confidential reports to the Editor). Please note this decision does not guarantee eventual acceptance.

Please submit a copy of your revised paper before 15-Sep-2019. Please note that the revision deadline will expire at 00.00am on this date. If we do not hear from you within this time then it will be assumed that the paper has been withdrawn. In exceptional circumstances, extensions may be possible if agreed with the Editorial Office in advance. We do not allow multiple rounds of revision so we urge you to make every effort to fully address all of the comments at this stage. If deemed necessary by the Editors, your manuscript will be sent back to one or more of the original reviewers for assessment. If the original reviewers are not available, we may invite new reviewers.

- Data accessibility

<http://datadryad.org/submit?journalID=RSOS&manu=RSOS-190490>

- Competing interests

- Authors' contributions

- Acknowledgements

- Funding statement

on behalf of Jon Blundy (Subject Editor)
openscience@royalsociety.org

Associate Editor's comments:

Thank you for submitting this manuscript to the journal. Please ensure that you fully respond to and incorporate the changes required by the reviewers, as your revised manuscript will be sent to them to review once more.

Reviewers' Comments to Author:

Reviewer: 1

The paper contributes to better understanding of the impact of farming practices to surface water quality in small agriculture dominated catchments.

Page 1, line 57 and page 3 lines 16-17: Perhaps "...13-15 kg hm⁻² and 7-17 kg hm⁻²..." and "declined to 74-112 kg hm⁻².."?

P 3, lines 21-23 : N harvested from the two catchments is not exactly similar (96 and 105 kg hm⁻²).

P3, line 29: "normal period" "normal temperature"? long-term mean in 1961-1990? Long-term mean temperature in 1961-1990?

P3, line 38: Please provide more info about the coefficients. What are they?

P3, lines: 42-23: what does "similar variations" mean? Similar in-stream concentrations or between the streams?

P 3, lines 58-60: The results of N loss in the catchments are provided already in the beginning of chapter 4.4.

P3, last row: perhaps 11.3 kg hm⁻²?

P4: It would be more logical if the chapter 4.6 (monthly concentrations) follows the chapter 4.3 (annual concentrations.)

P4: Something is wrong with numbering of figures 5, 6 and 7. Figure 5 is about monthly average values for TN and nitrate-N concentrations and not about N runoff. Fig 6 is about N loss and not N concentrations and Fig 7 is about correlation between concentrations and water runoff and N loss.

P4, lines 40-41. How to understand provided indexes. Index is a relationship between two numbers. Please add some explanation.

P4, line 49: "...and more intensive agricultural production with annual crops." What does this mean? Please consider revising of the sentence.

P 4, lines 50-52: Please add actual data about N surplus in the results part.

P4, lines 54-55: Please remove the sentence. You already described the used method to calculate N loss in the methods chapter?

P4, line 56: Something is missing here. "The study results showed that...."?

P 4, line 58: "mean annual concentration"?

P5, line 3: How to understand that? N use efficiency and N losses are largely defined by cropping and tillage systems?

P5, lines 5-6: "In this study.....". This information is provided already under the site descriptions.

P5, lines 9-10: dominating soil type..?

P5, line 18: "...increased soil tillage..." . Increased compared to some previous periods? Any data?

P5, lines 20-24: How to understand that? Fall is a typical season without plant growth? Temperatures are increasing? Lower summer temperatures are responsible for more mild climate? Probably not "climate" but "weather conditions". Climate is defined as long-term (> 30 years) average state of weather condition.

P5, line 30: Should be Povilaitis et al. (33).

P5, lines 39-41: OK. Biochar could prevent N loss. Whether biochar is actually applied in the Naurstad catchment? How does this sentence relates to your work? Please provide some more explanation.

P5, lines 43-44: One of the previous studies revealed that "Nitrogen leaching loss was influenced much more by the amount of organic matter..." How do you know this the case in two studied catchments? "...was probably influenced...?"

P5, line 53: "... but these properties were not present in the Naurstad catchment". Is it really so? N concentrations were rather stable in the Naurstad catchment (e.g. Figure 5) but seasonal N loss varied a lot?

P 6: Please check names of the authors for reference 33.

P 8: figure captions. figures 1-4 in 1994-2016, in 1994-2010.

Fig 7 includes two figures (7A and 7B). Which one is A and B? Axis titles and units are missing?

The manuscript can be accepted for publication if these concerns are properly addressed.

Reviewer: 2

Comments to the Author(s)

The paper is generally well structured and written. However, I have some comments:

What was the frequency of water quality tests?

Line 58 Total Nitrogen is the sum of nitrate (NO₃), nitrite (NO₂), organic nitrogen and ammonia.

The mathematical formula requires a comment, maybe the Authors should change the symbols.

Figure 7 has no descriptions (incorrectly created pdf), I do not know if it contains relevant data and matching measures.

Table 1 - main crops - GRAIN? Maybe cereals?

Table 2 - please remove "normal temperature" and "normal precipitation". Use "mean annual temperature" annual precipitation, Long-term - please define this period 1994-2016.

It was mentioned about the correlation between runoff and nitrogen losses (Figure 5A does not exist in the pdf file), what about rainfall-runoff correlation (Figure 2 does not contain this information)? Nitrogen transfer pathways to surface waters should be clearly described/indicated. Section 4.8 is unclear.

Please, emphasize the contribution of your research to the literature.

Reviewer: 3

Comments to the Author(s)

The paper is interesting reporting on long-term N losses from nordic agricultural catchments. It provides valuable information for researchers and stakeholders. However, a minor revision is recommended.

Regarding the measures to mitigate N losses there is a gap which is world-wide accepted. I would recommend to shortly discuss the role of riparian buffer zones and constructed wetlands.

See e.g.: Mander, Ü., Tournebize, J., Tonderski, K., Verhoeven, J.T.A., Mitsch, J.W., 2017.

Planning and establishment principles for constructed wetlands and riparian buffer zones in agricultural catchments. Ecological Engineering 103B, 296-300.

Author's Response to Decision Letter for (RSOS-190490.R0)

See Appendix A.

RSOS-190490.R1 (Revision)

Review form: Reviewer 1

Is the manuscript scientifically sound in its present form?

Yes

Are the interpretations and conclusions justified by the results?

Yes

Is the language acceptable?

Yes

Do you have any ethical concerns with this paper?

No

Have you any concerns about statistical analyses in this paper?

No

Recommendation?

Accept as is

Comments to the Author(s)

Raised questions are properly answered and all proposals for the revision considered.

Review form: Reviewer 2

Is the manuscript scientifically sound in its present form?

Yes

Are the interpretations and conclusions justified by the results?

Yes

Is the language acceptable?

Yes

Do you have any ethical concerns with this paper?

No

Have you any concerns about statistical analyses in this paper?

No

Recommendation?

Accept as is

Comments to the Author(s)

The article has been improved according to the reviewer's comments.

Decision letter (RSOS-190490.R1)

06-Nov-2019

Dear Dr Chen:

On behalf of the Editors, I am pleased to inform you that your Manuscript RSOS-190490.R1

entitled "Nitrogen losses from two contrasting agricultural catchments in Norway" has been accepted for publication in Royal Society Open Science subject to minor revision in accordance with the referee suggestions. Please find the referees' comments at the end of this email.

The reviewers and Subject Editor have recommended publication, but also suggest some minor revisions to your manuscript. Therefore, I invite you to respond to the comments and revise your manuscript.

- Ethics statement

- Data accessibility

If you wish to submit your supporting data or code to Dryad (<http://datadryad.org/>), or modify your current submission to dryad, please use the following link:
<http://datadryad.org/submit?journalID=RSOS&manu=RSOS-190490.R1>

- Competing interests

- Authors' contributions

- Acknowledgements

- Funding statement

Because the schedule for publication is very tight, it is a condition of publication that you submit the revised version of your manuscript before 15-Nov-2019. Please note that the revision deadline will expire at 00.00am on this date. If you do not think you will be able to meet this date please let me know immediately.

on behalf of Prof Jon Blundy (Subject Editor)
openscience@royalsociety.org

Associate Editor Comments to Author:

Your manuscript is essentially ready for publication; however, it has been observed that a number of the files etc. in the supporting data include non-English names for variables. Royal Society Open Science publishes in English, so please can you amend these variables to their English equivalents?

Reviewer comments to Author:

Reviewer: 1

Comments to the Author(s)

Raised questions are properly answered and all proposals for the revision considered.

Reviewer: 2

Comments to the Author(s)

The article has been improved according to the reviewer's comments.

Author's Response to Decision Letter for (RSOS-190490.R1)

See Appendix B.

Decision letter (RSOS-190490.R2)

14-Nov-2019

Dear Dr Chen,

It is a pleasure to accept your manuscript entitled "Nitrogen losses from two contrasting agricultural catchments in Norway" in its current form for publication in Royal Society Open Science. The comments of the reviewer(s) who reviewed your manuscript are included at the foot of this letter.

on behalf of Prof Jon Blundy (Subject Editor)
openscience@royalsociety.org

Appendix A

1 Associate Editor's comments:

Thank you for submitting this manuscript to the journal. Please ensure that you fully respond to
and incorporate the changes required by the reviewers, as your revised manuscript will be sent to
them to review once more.

Reviewers' Comments to Author:

Reviewer: 1

The paper contributes to better understanding of the impact of farming practices to surface water
quality in small agriculture dominated catchments.

Page 1, line 57 and page 3 lines 16-17: Perhaps "...13-15 kg hm⁻² and 7-17 kg hm⁻²..." and
"declined to 74-112 kg hm⁻².."?

**Answer:** Thank you for this feedback.

P 3, lines 21-23 : N harvested from the two catchments is not exactly similar (96 and 105 kg hm⁻²
2).

**Answer:** Yes, you are right. These values are not exactly similar according to the average values,
so we revised this phrase to "fluctuated between 74 kg hm⁻² and 116 kg hm⁻²".

P3, line 29: "normal period" "normal temperature"? long-term mean in 1961-1990? Long-term
mean temperature in 1961-1990?

**Answer:** Thank you. These are not long-term mean values but instead refer to the standard 30-
25 year normal period (1961-1990). We have made changes in the text to specify that these are 30-
26 year normal values for precipitation and temperature.

P3, line 38: Please provide more info about the coefficients. What are they?

**Answer:** We calculated the correlation between runoff and rainfall in two catchments to determine
whether rainfall was the main reason for the occurrence of runoff. However, we believe that this
information was not enough to explain this problem, so we deleted this sentence.

P3, lines: 42-23: what does "similar variations" mean? Similar in-stream concentrations or
between the streams?

**Answer:** We meant that the fluctuations of TN and the nitrate-N concentrations among these
monitoring years were similar between two streams. We change "variations" to
"fluctuations".

P 3, lines 58-60: The results of N loss in the catchments are provided already in the beginning of
chapter 4.4.

**Answer:** We apologize for the mistake and deleted the sentence "Similar results for nitrate-N
loss in the two catchments indicated that the average annual nitrate-N losses were 32.2 kg
hm⁻² in the Skuterud catchment and 8.7 kg hm⁻² in the Naurstad catchment."

P3, last row: perhaps 11.3 kg hm⁻²?

**Answer:** Yes, thank you very much. It should be 11.3 kg hm⁻².

P4: It would be more logical if the chapter 4.6 (monthly concentrations) follows the chapter 4.3
(annual concentrations.)

**Answer:** Yes, we agree with your comments, and we have changed the order of these chapters.

P4: Something is wrong with numbering of figures 5, 6 and 7. Figure 5 is about monthly average
values for TN and nitrate-N concentrations and not about N runoff. Fig 6 is about N loss and not N
concentrations and Fig 7 is about correlation between concentrations and water runoff and N loss.

**Answer:** We apologize for the incorrect order. The figures have been rearranged, and Figure 5
now refers to N loss. Figure 6 refer to the monthly average values for TN and nitrate-N
concentrations. Figure 7 refers to the correlation between the concentrations and water runoff and
N loss.

P4, lines 40-41. How to understand provided indexes. Index is a relationship between two
numbers. Please add some explanation.

**Answer:** The index is a correlation coefficient used to identify the effect of differences in runoff
and nitrogen concentrations on nitrogen loss based on annual data for 23 years. We have changed
the text and refer to coefficients instead of the index.

P4, line 49: “.....and more intensive agricultural production with annual crops.” What does this
mean? Please consider revising of the sentence.

**Answer:** This sentence means that there are more agricultural areas in the Skuterud catchment
than grass planting areas in the Naurstad catchment

We have changed to “In comparison with the Naurstad catchment, which is dominated by
grasslands, the Skuterud catchment in southern Norway has a highly intensive agricultural
cropping system with mainly cereal production.”

P 4, lines 50-52: Please add actual data about N surplus in the results part.

**Answer:** Yes, thank you for this feedback.

“The average N surplus was 66 kg hm⁻² in the Skuterud catchment and 7 kg hm⁻² in the
Naurstad catchment over the monitoring period.....”

P4, lines 54-55: Please remove the sentence. You already described the used method to calculate
N loss in the methods chapter?

**Answer:** We have deleted this sentence.

P4, line 56: Something is missing here. "The study results showed that....."?

**Answer:** Yes, thank you for this feedback.

P 4, line 58: “mean annual concentration”?

**Answer:** We have changed this phrase to “annual mean concentration”, thank you.

P5, line 3: How to understand that? N use efficiency and N losses are largely defined by cropping
and tillage systems?

**Answer:** The text has been revised to improve the understanding
N use efficiency was calculated according to the nitrogen harvest (uptake by plants) / nitrogen
application *100, which means the cropping system plays a decisive role in this process. Of
course, the nutrients in soil that are available to plants also affect the plant uptake amounts. The
importance of the cropping system has been shown in reference 22.

P5, lines 5-6: "In this study.....". This information is provided already under the site
descriptions.

**Answer:** We have removed this sentence.

P5, lines 9-10: dominating soil type..?

**Answer:** Yes, you are right. "Silty clay is the dominant soil type in the Skuterud catchment, and
peat is the dominant soil type in Naurstad."

P5, line 18: "...increased soil tillage..." . Increased compared to some previous periods? Any
data?

**Answer:** Increased is in comparison with the values for no tillage in the Naurstad catchment. We
change "increased" to "more".

P5, lines 20-24: How to understand that? Fall is a typical season without plant growth?
Temperatures are increasing? Lower summer temperatures are responsible for more mild climate?
Probably not "climate" but "weather conditions". Climate is defined as long-term (> 30 years)
average state of weather condition.

**Answer:** Yes, fall is the typical period without plant growth in the areas dominated by spring cereals.
Thank you for your comments, the text on mild climate has been deleted.

P5, line 30: Should be Povilaitis et al. (33).

**Answer:** Thank you for this feedback.

P5, lines 39-41: OK. Biochar could prevent N loss. Whether biochar is actually applied in the
Naurstad catchment? How does this sentence relates to your work? Please provide some more
explanation.

**Answer:** No, biochar was not used in the Naurstad catchment; thus, we removed this information.

P5, lines 43-44: One of the previous studies revealed that "Nitrogen leaching loss was influenced
much more by the amount of organic matter..." How do you know this the case in two studied
catchments? "...was probably influenced...?"

**Answer:** There are more livestock and grass planting areas in the Naurstad catchment than in
the other catchment, and we believed they were the main reason for the high rate of organic
N loss to TN loss.

“Nitrogen leaching loss was influenced much more by the amount of organic matter
accumulated in the soil in previous years than by the mineral N fertilization of the current
crop [34]. In this study, grass residue and higher livestock density probably resulted in the
higher rate of organic N loss to TN loss in the Naurstad catchment than in the Skuterud
catchment.”

P5, line 53: “.... but these properties were not present in the Naurstad catchment”. Is it really so?
N concentrations were rather stable in the Naurstad catchment (e.g. Figure 5) but seasonal N loss
varied a lot?

**Answer:** Yes, you are right, we have changed this sentence to “Nitrogen concentrations in the
Skuterud catchment showed seasonal variations with the highest concentrations in spring
and autumn, whereas the nitrogen concentrations in the Naurstad catchment did not vary
much over the year.”

P 6: Please check names of the authors for reference 33.

**Answer:** Yes, there was a missing letter.

Povilaitis A , Šileika, Antanas, Deelstra J , et al. Nitrogen losses from small agricultural
catchments in Lithuania[J]. Agriculture, Ecosystems & Environment, 2014, 198:54-64.

P 8: figure captions. figures 1-4 in 1994-2016, in 1994-2010.

**Answer:** It should be 1994-2016.

Fig 7 includes two figures (7A and 7B). Which one is A and B? Axis titles and units are missing?

**Answer:** Fig. 7 has been changed to Fig. 6, which has been labelled as Fig 6A and Fig 6B. We
have redrawn Fig. 6 according to annual runoff, total nitrogen and nitrate nitrogen loss, as follows.

The manuscript can be accepted for publication if these concerns are properly addressed.

**Answer:** Thank you very much for your valuable comments, we have revised our manuscript and
answered your comments one by one, thanks again.

Reviewer: 2

Comments to the Author(s)

The paper is generally well structured and written. However, I have some comments:

What was the frequency of water quality tests?

**Answer:** Water sampling was implemented by a data logger when a fixed predetermined
volume of water passed the measuring station, while a small water sample was taken and
added to a container that was kept in a refrigerator in the monitoring station. Samples were
collected for analysis every fourteen days.

Line 58 Total Nitrogen is the sum of nitrate (NO₃), nitrite (NO₂), organic nitrogen and ammonia.
The mathematical formula requires a comment, maybe the Authors should change the symbols.

**Answer:** We changed the formula as follows,

$ON = TN \text{ (kg hm}^{-2}\text{)} - \text{Nitrate-N.}$

ON-Organic N, means nitrogen lost from runoff water in organic nitrogen form kg hm⁻²

TN-Total N, means nitrogen lost from runoff water in total nitrogen form, kg hm⁻²

NN-Nitrate-N, means nitrogen lost from runoff water in nitrate nitrogen form, kg hm⁻²

Figure 7 has no descriptions (incorrectly created pdf), I do not know if it contains relevant data
and matching measures.

**Answer:** We apologize for the mistake. Figure 7 is as follows.

Table 1 - main crops - GRAIN? Maybe cereals?

**Answer:** The Skuterud catchment is dominated by cereals and the Naustad catchment is
dominated by grassland. We use cereal instead of grain throughout the manuscript

Table 2 – please remove “normal temperature” and “normal precipitation”. Use “mean annual
temperature” annual precipitation, Long-term – please define this period 1994-2016.

**Answer:** Yes, thank you for this feedback. We changed “normal temperature” and “normal
precipitation” to “mean annual temperature for the monitoring period and for the 30-year
normal period (1961-1990)” and “mean annual precipitation for the monitoring period and for the 30-year
normal period (1961-1990)”,

It was mentioned about the correlation between runoff and nitrogen losses (Figure 5A does not
exist in the pdf file), what about rainfall-runoff correlation (Figure 2 does not contain this
information)?

**Answer:** We apologize, as the numbers in Figure 5 and Figure 7 were wrong.
The correlation between rainfall and runoff was calculated with SPSS software using data in
Figure 2, and the correlation coefficient is shown in the manuscript but not in the figure. We
believe that this correlation was not enough to explain the contribution, so we deleted this
sentence.

Nitrogen transfer pathways to surface waters should be clearly described/indicated.

**Answer:** In the background section, we have included a text on nitrogen transfer: “Increases in N
fertilizer inputs and the N surplus result in the leaching of N from agricultural soils, and N losses
at the catchment scale result in environmental problems in streams, river, lakes and the sea.”

Section 4.8 is unclear.

**Answer:** In this section, we intended to identify the main contributor of N losses from the two
catchments. In this study, these two catchments have significantly different properties, including
cropping system, soil types, livestock densities, and agricultural areas, which resulted in different
nitrogen concentrations. The rainfall and temperature conditions resulted in different runoff
amounts. Therefore, we intended to find the main factor that determined the N loss in the two
catchments.

Please, emphasize the contribution of your research to the literature.

**Answer:** Our study intended to emphasize the effect of agriculture planting on nitrogen loss from
runoff water at the catchment scale.

- • The N surplus in agricultural production was ten times higher for the areas dominated by
cereal crops than for the areas dominated by grassland. The growing season for cereals is
much shorter than that for grassland, and the possible period for N uptake is
correspondingly shorter. Hence, the crops with longer growing seasons reduced the risk of
N loss to water compared to the crops with a short growing season.
- • The seasonal variations in nitrogen losses in both cereal crops and grassland are dominated
by high losses in spring and autumn. Special care should be taken to maintain efficient
uptake by crops during these periods. Increasing the crop cover for cereal production, e.g.,
by using catch crops, will reduce the N loss.
- • The N losses from mineral soils with cereal production are mainly in the form of nitrate-N
whereas for organic soil with grass-production, the organic N forms dominate due to low
nitrate-N loss.
- • The decreasing trend in N application and N-surplus during the monitoring period
contributed to decreased N loss.
- • The increase in runoff caused the increase in N loss from both production systems that
were studied.

44 Reviewer: 3

Comments to the Author(s)

The paper is interesting reporting on long-term N losses from nordic agricultural catchments. It
provides valuable information for researchers and stakeholders. However, a minor revision is
recommended.

Regarding the measures to mitigate N losses there is a gap which is world-wide accepted. I would
recommend to shortly discuss the role of riparian buffer zones and constructed wetlands.

See e.g.: Mander, Ü., Tournebize, J., Tonderski, K., Verhoeven, J.T.A., Mitsch, J.W., 2017.

Planning and establishment principles for constructed wetlands and riparian buffer zones in
agricultural catchments. Ecological Engineering 103B, 296-300.

**Answer:** We have included a discussion on buffer zones and constructed wetlands in the

Discussion section.

Appendix B

Dear editors and reviewers,

Thank you very much for your comments to improve our manuscript. And our manuscript was accepted by journal of Royal Society Open Science

About comments from editors and reviewers we responded as follow:

Associate Editor Comments to Author:

Your manuscript is essentially ready for publication; however, it has been observed that a number of the files etc. in the supporting data include non-English names for variables. Royal Society Open Science publishes in English, so please can you amend these variables to their English equivalents?

We have done the suggested changes.

Reviewer comments to Author:

Reviewer: 1

Comments to the Author(s)

Raised questions are properly answered and all proposals for the revision considered.

Thanks a lot.

Reviewer: 2

Comments to the Author(s)

The article has been improved according to the reviewer's comments.

Thanks a lot.